# hTERT and SV40LgT Renal Cell Lines Adjust Their Transcriptional Responses After Copy Number Changes from the Parent Proximal Tubule Cells

**DOI:** 10.3390/ijms26083607

**Published:** 2025-04-11

**Authors:** Bruce Alex Merrick, Ashley M. Brooks, Julie F. Foley, Negin P. Martin, Rick D. Fannin, Wesley Gladwell, Kevin E. Gerrish

**Affiliations:** 1Mechanistic Toxicology Branch, Division of Translational Toxicology, National Institute of Environmental Health Sciences, Research Triangle Park, Durham, NC 27709, USA; foley1@niehs.nih.gov; 2Biostatistics and Computational Biology Branch, Integrative Bioinformatics Support Group, Division of Intramural Research, National Institute of Environmental Health Sciences, Research Triangle Park, Durham, NC 27709, USA; ashley.brooks@nih.gov; 3Viral Vector Core, Neurobiology Laboratory, Division of Intramural Research, National Institute of Environmental Health Sciences, Research Triangle Park, Durham, NC 27709, USA; martin12@niehs.nih.gov; 4Molecular Genomics Core Laboratory, Division of Intramural Research, National Institute of Environmental Health Sciences, Research Triangle Park, Durham, NC 27709, USA; rick.fannin@gmail.com (R.D.F.); gladwell@niehs.nih.gov (W.G.); gerrish@niehs.nih.gov (K.E.G.)

**Keywords:** kidney, proximal tubule, immortalization, transcriptomics, copy number variation

## Abstract

Primary mouse renal proximal tubule epithelial cells (moRPTECs) were immortalized by lentivirus transduction to create hTERT or SV40LgT (LgT) cell lines. Prior work showed a more pronounced injury and repair response in LgT versus hTERT cells after chemical challenge. We hypothesized that unique genomic changes occurred after immortalization, altering critical genes and pathways. RNA-seq profiling and whole-genome sequencing (WGS) of parent, hTERT, and LgT cells showed that 92.5% of the annotated transcripts were shared, suggesting a conserved proximal tubule expression pattern. However, the cell lines exhibited unique transcriptomic and genomic profiles different from the parent cells. Three transcript classes were quite relevant for chemical challenge response—Cyps, ion channels, and metabolic transporters—each important for renal function. A pathway analysis of the hTERT cells suggested alterations in intermediary and energy metabolism. LgT cells exhibited pathway activation in cell cycle and DNA repair that was consistent with replication stress. Genomic karyotyping by combining WGS and RNA-seq data showed increased gene copy numbers in chromosome 5 for LgT cells, while hTERT cells displayed gene copy losses in chromosomes 4 and 9. These data suggest that the exaggerated transcriptional responses of LgT cells versus hTERT cells result from differences in gene copy numbers, replication stress, and the unique selection processes underlying LgT or hTERT immortalization.

## 1. Introduction

Renal proximal tubules play a critical role in kidney function, accounting for approximately 70% of fluid reabsorption, uptake of electrolytes, intermediary metabolites, and nutrients, as well as functions in acid–base homeostasis, hormone production, and secretion of waste products [1]. Isolation of primary renal proximal tubule epithelial cells (RPTECs) typically involves collagenase digestion, centrifugation, and sieving, often requiring further purification by Percoll gradients [2,3,4] or flow cytometry [5]. Primary cells are valued for their biochemical characteristics similar to in vivo tissue. Even so, primary proximal tubule cells have a limited life span in culture, with a continual need for isolation to conduct multiple experiments that may entail interlaboratory reproducibility issues.

Cell immortalization provides a means to maintain many of the biochemical and phenotypic properties of primary cells for comparability of experiments and distribution among different laboratories [6]. Two common immortalization methods are viral expression vectors for human telomerase reverse transcriptase (hTERT) or SV40 Large T antigen (SV40LgT). hTERT immortalizes cells by maintaining telomere length to prevent senescence [7], while SV40LgT binds to and inactivates tumor suppressors, such as Rb and Trp53 proteins [8]. Some researchers prefer hTERT immortalization to other methods citing phenotypic traits similar to primary cells and with less karyotypic changes or DNA injury [9,10,11,12,13]. Others have noted that hTERT expression can produce genomic reorganization and instability depending on the cell type, genetic background, and donor age [14,15,16].

Immortalization of renal proximal tubules has been demonstrated to successfully retain many biochemical activities of this multifunctional cell type. Human RPTECs/TERT1 cells have wide use in the study of the pharmaceutical effects of drug and ion transporter activities [17,18,19], in nephrotoxicity screening assays [20,21,22,23], and in describing physiological functions of proximal tubules, such as membrane voltage, oxygen tension, and 3D tubule formation with polarity features [24,25,26,27]. Further, RPTEC/TERT1 immortalized cells are frequently used as a normal cell type in studies on renal cancer cell development and chemotherapeutic treatments [28,29,30]. The inclusion of continuous or pulsatile flow, polarity, microstructures (e.g., microvilli and basement membrane formation), multicellular organoids, and kidney-on-a-chip are among the many recent features being introduced by research consortia into sophisticated, microphysiological models using immortalized renal cell types [27,31,32,33,34]. Further, microfluidic systems aim to better emulate high flow organs like the kidneys by applying flow and fluid shear stresses to better replicate the cellular phenotype and function that occur under physiological conditions [35].

Immortalized mouse RPTEC cells lines by SV40LgT have found wide use in cell biology and pharmaceutical investigations [36]. For example, MKCC cells from proximal tubules of SV40 transgenic mice were used to study Na^+^/H^+^ antiporters reflecting a polarized phenotype in these immortalized cells [37]. Similarly, PKSV-PCT and PKSV-PR cells [38] demonstrated a polarized phenotype developing an apical brush border expressing villin and aminopeptidase N. Later, PKSV-PCT cells were proposed as a screen for the adverse renal effects of pharmacologic agents when N-acetyl-beta-D-glucosaminidase (NAG) and acid phosphatase (ACP) leakage was observed after gentamicin or chloroquine exposure [39]. The TKPTS cell line, developed from 8Tg(SV40E)Bri7 mouse proximal tubules, was used to study PGP-mediated drug transport [40]. tsMPT cells, created with temperature-sensitive SV40LgT cells [41], demonstrated expression of semaphorins and their neuropilin receptors, important for directing kidney vascularization [42]. mProx cells were immortalized by transfection of SV40LgT antigen into microdissected C57Bl/6 mouse proximal tubule cells to show ERK pathway involvement in albumin-induced MCP-1 expression [43].

We recently created two new mouse proximal tubule cell lines from primary proximal tubule cells from CD-1 mice by immortalization with lentivirus vectors containing hTERT or SV40LgT (LgT) [44]. Immortalized cells were compared by RNA-seq expression profiling after repeated, subcytotoxic concentrations of cisplatin (CisPt) and aflatoxin B1 (AFB1). Only LgT cells transcriptionally responded to AFB1 exposure, indicating mild toxicity while both cell types had a robust expression response to CisPt over a 3–96 h timeframe, showing concurrent damage and repair. Though individual gene responses to CisPt showed some variation between these two cell lines, the activated pathways and controlling genes converged over time into a coordinated program consistent with the concept of a ‘renal repair transcriptome’ [45]. Pathway analysis comparing hTERT and LgT cells expression patterns to these nephrotoxicants revealed shared responses in CREB signaling in neurons for cellular plasticity; GPCR activation for signal transduction of hormonal, cytokine, and chemokine ligands; tissue remodeling, inflammation, and immune cell recruitment; and regulation of growth, differentiation, and developmental processes. However, it was notable that LgT cells had greater transcriptional responsiveness to chemical challenge than hTERT cells reflected in a high fold change in affected genes. In the current work, we hypothesize that each immortalization process creates an altered transcriptional and genomic landscape that underlies their phenotypic responses to nephrotoxicants. Therefore, we conducted RNA-seq expression and whole genome sequencing analysis in hTERT and LgT mouse RPTECs (moRPTEC) for comparison to that from normal parent proximal tubule cells to uncover the altered genes and pathways responsible for the differing phenotypic properties in these two cell lines.

## 2. Results

### 2.1. RNA-Seq Results

Primary RPTECs and the two moRPTEC cell lines, hTERT and LgT, were cultured to near confluency in 6-well plates prior to the isolation of RNA, library preparation, and sequencing. Each cell type involved six replicates. The results of the RNA-seq are presented in Appendix A. Each tab contains data for 55,401 Ensembl transcripts as a modified DESeq2 output that includes the mouse Ensembl ID, gene symbol and name, RefSeq and Entrez IDs, base mean counts (BMCs), fold change compared to primary moRPTEC controls as log_2_ or log_10_, standard error (IfcSE), and *p*-values. Up- or downregulated calls for differentially expressed genes (DEGs) compared to primary cells were made based on a ≥2-fold change at an adjusted *p*-value of *p* ≤ 0.05 (pAdj ≤ 0.05).

Table 1 shows over 23,000 Ensembl transcripts were detected at a ≥5 BMC threshold of reads per transcript in hTERT and LgT cells. The total number of transcripts in each cell line that were DEGs from parent proximal tubule cells was generally equivalent (5740 DEGs for LgT and 6426 DEGs for hTERT). Another 9998 to 12,333 Ensembl transcripts were classified as ‘No Change’ since they did not meet the ≥2× fold change criteria for differential expression but met the ≥5 BMCs as a threshold set for measurable expression. We also considered Ensembl transcripts that had functional annotations after the removal of the ‘NA’ Ensembl transcripts. Annotated transcripts at ≥5 BMCs numbered 10,051 transcripts (Appendix A) of which >92% of expressed genes were common among parent, hTERT and LgT cells. This observation suggested a high degree of expression similarity that might be expected among renal proximal tubule cells of either primary or immortalized origin. We then explored the hypothesis that differential expression might relate to the activation of critical pathways and the behavior of each cell line that we observed in the response to a toxicant challenge.

In our previous work, we observed that only LgT cells (not hTERT) showed a transcriptional response to aflatoxin B1 (AFB1), indicating damage without overt cytotoxicity, and showed a more rapid rise in transcriptional changes to CisPt compared to hTERT cells [44]. We did not test primary proximal tubule cells’ response to nephrotoxicants in this prior study. However, we wanted to continue probing how these two cell lines compared at the transcriptional and genomic levels to those of primary proximal tubule cells as a means to explain these differing phenotypic responses to the two cytotoxicants, aflatoxin B1 (AFB1) and cisplatin (CisPt).

### 2.2. Venn Diagram and Pathway Analysis

One approach to answering this question is a Venn diagram analysis, using normal proximal tubule transcripts as comparators to the two cell lines. The differentially expressed genes (DEGs) of each cell line versus the normal parent cells can parse both common and unique DEGs for each line. We found each cell line had a slightly different transcriptome compared to the parent cells. There were just 265 upregulated DEGs in common for the two cell lines but each cell line had a set of unique DEGs compared to the parent cells. There were 800 unique upregulated DEGs for hTERT cells and 1262 unique upregulated DEGs for LgT cells.

Pathway analysis was used to derive further meaning from the Venn diagram data. Appendix A contain the pathways associated with the shared genes among the parent and the two cell lines. Figure 1 shows the top 10 shared canonical pathways for hTERT and LgT. Selected DEGs and fold changes are indicated for the top scoring pathways, such as neurovascular coupling, hypercytokinema, semaphorin neural repulsion, and pyrimidine synthesis. There was general agreement in the ranking of common pathways for the two cell lines, although variations in the fold-change intensity accounted for some differences in pathway ranking. Notable increases in expression occurred in these cells for the big potassium BK channel, Lrrc26 [46], from 26.1× to 53.6× fold; for the autocrine growth factor, Areg [47], from 24.4× to 2.4× fold; and for the extracellular matrix proteoglycan, Acan, from 19.8× to 20.1× fold, which coordinates with the semaphorin, Sema3 [48]. Careful monitoring of ATP levels is performed by the ectonucleotidase family (Entpd2,3,8) which regulates neurovascular and immune functions in kidney [49].

From the Venn diagram analysis, 800 DEGs unique to hTERT (Appendix A) and 1262 DEGs unique to LgT (Appendix A) cells were found when compared to primary proximal tubule cell expression. The top 20 pathways are shown in Figure 2 for hTERT cells and Figure 3 for LgT cells. The pathway enrichment for each unique gene set suggests different patterns of pathway activation in each cell line. The hTERT cell pathways (Figure 2) generally reflected alterations in intermediary and synaptic transmitter metabolism. For example, increases in aldehyde dehydrogenases are relevant to the degradation of serotonin, ethanol, dopamine, tryptophan, and noradrenaline. Activation of the GABAergic system was supported by increased Gad2 (enzyme for conversion of glutamic acid to GABA) that is interconnected with glycine biosynthesis. Relatedly, axon guidance pathways regulate the synaptogenesis of neurotransmitters and intravesicular transport. Other intermediary systems of interest in hTERT cells involve GSH-transferase enzymes including the PXR and CAR signaling pathways, as well as GSH conjugation reactions. Gene expression and pathway changes generally reflect adjustments in intermediary and bioenergetic metabolism in hTERT cells compared to parent cells.

A pathway analysis of upregulated DEGs in LgT cells exhibited a somewhat different pattern of activation than hTERT cells (Appendix A). Cell cycle regulation and DNA repair pathways were generally engaged in LgT cells, as shown in Figure 3. Although DEGs increase, those populating activated pathways were generally in the 2× to 3×-fold change range, and the regularity of the cell cycle and DNA repair pathway activation was apparent. Notable LgT transcript changes compared to parent proximal tubules included a 23×-fold increase in STAG3, a cohesion protein involved in sister chromatid generation during the cell cycle [50]; a 15×-fold increase in the cell cycle regulator Cdkn2a (p16^Ink4a^) [51]; and increases in Brca1 at 2×-fold and Brca2 at 5×-fold as mediators of ongoing chromosomal repair [52]. These events represent portions of the replication stress response involving changes in the cell cycle, DNA repair, and replication fork integrity, as recently reviewed [53].

### 2.3. Differential Expression of Genes in Development, IFN-Response, and Cdk Inhibition

We also considered DEGs in gene groups not appearing in the pathway analysis for LgT and hTERT cells. These genes can be grouped by functional class (Appendix A) and included genes involved in embryonic and organ development, interferon response genes, and cdk inhibition. For LgT, Hox13a,b,d developmental genes increased from 2× to 41×-fold compared to parent cells, while several homeobox transcripts increased in both hTERT and LgT cells, such as Cdx2, Esx1, Hopx, and Nkx2–3. Several Fox genes were upregulated in both cell lines, including Foxg1 at 75×-fold in LgT cells and at 36×-fold in hTERT cells. Forkhead box (Fox) transcription factors are involved in multiple early developmental processes, differentiation, and tissue maintenance [54]. Finally, Wt1 is a transcription factor critical for normal embryonic kidney development and in adulthood; it, generally, acts as a tumor suppressor [55]. Here, we found its expression to be greatly reduced, at −99×-fold in LgT cells and at −285×-fold in hTERT cells compared to parent cells.

Interferon-beta (Infb1) was upregulated 35×-fold in hTERT and 12×-fold in LgT cells (Appendix A). Infb1 is a potent regulator of many downstream interferon-inducible genes [56], including the Oas (2′,5′-oligoadenylate synthetase) and Mx (MX dynamin-like GTPase I) gene families. Even though there was a higher fold increase in Infb1 for hTERT compared to LgT, a higher number of induced interferon-responsive genes occurred in LgT cells.

Cdk inhibitor genes regulate the cell cycle and checkpoints for mitosis and proliferation, and their inactivations are often involved in immortalization [57]. Cdkn2a (p16^Ink4a^) was unaffected in hTERT cells compared to parent proximal tubule cells, while Cdkn1c (p57^Kip2^) was greatly downregulated by 20×-fold, which may be responsible for immortalization in hTERT cells [58]. In contrast, we observed that Cdkn2a was upregulated by 14-fold in LgT cells. Only minor changes in expression compared to parent cells occurred for other Cdk inhibitors in either LgT or hTERT cells. While many of these Cdk inhibitor genes are largely controlled by changes in Trp53 expression, we note that Trp53 expression was essentially unchanged in both hTERT and LgT cells.

### 2.4. Differential Expression of Three Gene Classes Underlying Renal Pharmacodynamics

We further examined differences in the expression of important gene classes that underlie kidney function and are relevant for RPTECs’ investigational use in the study of disease and treatment. Three critical gene classes that affect pharmacology, therapeutics, and toxicology are small-molecule transporters, ion channels, and cytochrome P450 enzymes, since their expression impacts xenobiotic elimination or bioactivation to reactive intermediates. Over 550 Ensembl transcripts were combined to represent these three gene classes. To ensure we compared genes with adequate expression, we used a mean threshold of ≥5 BMCs for each transcript from *n* = 6 independent samples/group (see Appendix A).

### 2.5. Cytochrome P450s and Detoxification Enzymes

Firstly, cytochrome P450 enzymes (Cyp’s) are important for biotransformation of chemicals and pharmaceuticals. In all, 123 Cyp transcripts were measured by RNA-seq, but only 44 Cyp transcripts met the threshold criteria. A comparison of transcript expression was performed using Z-scores of the normalized BMCs, which account for the mean number of counts and standard deviation across cell types. Figure 4 shows Cyps clustered by Z-score into four transcript groups. Group ‘a’, composed of 19 transcripts, showed greater expression in parent cells, notably in arachidonic acid metabolism (Cyp2d9, Cyp2d12, Cyp2j5, Cyp2j6, Cyp2j9, and Cyp4a12a,b) and Ahr metabolism (Cyp1b1). Group ‘b’, composed of eight transcripts, had greater expression in parent cells and the LgT cell line than in hTERT cells, as well as in Cyp activity related to Ahr metabolism (Cyp1a1) and steroid biosynthesis (Cyp51). Group ‘c’, composed of 12 transcripts, showed that hTERT had more expression than in parent cells and LgT cells, especially for xenobiotic metabolism (Cyp3a13 and Cyp4b1) and retinoic acid metabolism (Cyp26a1 and Cyp26c1). Group ‘d’ showed that parent and hTERT cells had five transcripts with higher expression than LgT cells, such as Cyp2s1, which is involved in intermediary metabolism.

One of the study’s objectives was to understand the phenotypic difference between the two immortalized cell lines, since LgT cells exhibited AFB1 toxicity, while hTERT cells did not. The more important mouse cytochrome P450 (Cyp) bioactivation enzymes (e.g., Ahr and Cyp3) and detoxification enzyme families (e.g., epoxide hydrolase, GSH S-transferases, and aldo-keto reductases) were reviewed, and their basal-level transcript expression in counts by RNA-seq (upper panel), as well as differences in fold change compared to parent proximal tubule cells (lower panel), are shown in Figure 5. Transcripts for AFB1 metabolism in mice have been well studied and reported [59,60]. Fold changes for the most relevant metabolism transcripts are shown in bold in the lower panel. Cyp1a1 increased 2.3× fold in LgT cells, while Cyp3a13 also increased by 2.3× fold in hTERT cells. Other AFB1 bioactivating enzymes, such as Cyp1a2 and Cyp3a11, were not expressed, and the expression of Cyp1b1, although present, decreased in both cell lines compared to parent cells. Presuming AFB1 activation occurs in hTERT and LgT moRPTECs, these cells can deploy several detoxification enzyme pathways. Most notably, expression increased in hTERT cells, including several glutathione S-transferases (Gsta1a1, Gsta3, Gstk1, Gstm1, Gstm2, and Gstm3), epoxide hydrolases Ephx1 and Ephx2, and glutathione peroxidase (Gpx3). By contrast, these enzymes exhibited lower expression in LgT cells compared to parent cells. AFB1-specific aldo-keto reductase.

Akr7a5, was lower in both cell types, as were the redox enzymes catalase (Cat) and superoxide dismutases (Sod1, -2, and -3), as well as glutathione synthesis enzymes (Gsr and Gss).

### 2.6. Ion Channels

Ion channels play a critical role in renal function for maintaining Na^+^, K^+^, Cl^−^, and Ca^2+^ ion concentrations at homeostatic levels throughout the body by retaining or excreting ions as needed. For the 195 ion channels considered, Figure 6 shows that 150 transcripts met the threshold criteria of ≥5 BMCs. The transcripts were clustered into the following four groups: Group ‘a’, which has 62 parent transcripts with greater expression than LgT and hTERT cells; Group ‘b’, which has 40 parent and LgT cells with greater expression than hTERT cells; Group ‘c’, which has 28 hTERT transcripts with greater expression than parent and LgT cells; and Group ‘d’, which has 20 LgT transcripts with generally higher expression than parent and hTERT cells. Overall, the relatively reduced expression in 36 ion channel transcripts below 5 BMCs in hTERT cells suggests hTERT was the most affected cell type, followed by LgT cells with only 11 lower-expressing ion channels (Appendix A).

### 2.7. Renal Metabolic Transporters

Renal transporters play a key role in regulating the absorption of nutrients and excretion of waste or foreign substances such as drugs and chemicals. We looked for differential expression in 448 known transporters and found 364 transporter transcripts that met the threshold criteria of ≥5 BMCs (Appendix A). The transporter transcripts were segregated into seven clusters (Figure 7). A breakdown of the transcripts per cluster and the order of the greater and lower relative expressions is shown in Table 2. In addition, we describe the number of transcripts in each cluster with low expression at <5 BMCs for each cell line (despite the overall mean transcript expression of ≥5 BMCs for all three cell types). We observed that hTERT had 41 total transcripts in these seven clusters with low expression compared to 17 in LgT and only 5 low-expression transcripts in parent cells. Considering the relative order of each cell line from greater to lower expression, these data suggest a rebalancing of individual transporter expression within the hTERT and LgT transcriptomes after immortalization from parent cells.

Several classes of renal transporters are of particular interest to researchers in drug development, pharmacology, and toxicology, and are generally known by their protein identities, such as the OATs (organic anion transporters), OCTs (organic cation transporters), MDRs (multidrug resistance transporters), ENTs (equilibrative nucleoside transporters), among others [61,62]. We organized a panel of 60 transporters, as shown in Figure 8, by their protein ID and corresponding transcript ID to show each transcript’s relative expression from each cell type using the means of the BMCs from the RNA-seq data. After performing a statistical analysis of the BMC data (ANOVA and Newman–Keuls), we found 45 detectable transcripts (*p* < 0.05, in bold), which included at least one cell type with a group mean of ≥5 base mean counts. There were 15 (in grey font) transcript/proteins at low or undetectable expression levels of <5 BMCs, including two transporters for which we could not find a consensus transcript (grey font, red asterisk), precluding their analysis (see paragraph below). Highly expressed transporter classes in all cell types included equilibrative nucleotide transporters (e.g., Ent1,-2,-3), multidrug resistance transporters (e.g., Mdr1,-2, Mrp1,-3,-4,-5, Pgp, and Bcrp), organic cation transporters novel (e.g., Octn1,-2,-3), and glucose transporters (e.g., Glut1,-2,-9). However, other transporter classes exhibited relatively low expression, including Octs (e.g., Oct 2,-3), Oats (Oat1,-2,-3,-5,-6,-8), and the bile acid/steroid transporters OsTα,-β. Although transporter expression appeared somewhat higher in Parent cells, than hTERT and LgT cells, the overall transporter expression pattern in the two immortalized cell lines was well reflected with that of Parent cells.

Our review of renal transporters uncovered no consensus Ensembl transcripts for the following two transporters: mouse Oat4 and Oat7. The mus musculus Slc22a11 transcript (mOat4) has not been cloned and, so far, has only been classified as a predicted gene, Gm50435, by the Havana group at the Sanger Institute [63,64]. Similarly, there are discrepancies for a consistent transcript representing mouse Oat7 (mOat7) in the NCBI and Ensembl databases, each suggesting it could be related to Slc22a9 mammalian placental species orthologs or perhaps be the product of Slc22a19 or Slc22a21. As a result, no data points could be assigned to Oat4 or Oat7 in Figure 8 on the basis of the RNA-seq data.

### 2.8. Detection of Slc22a11 (mOat4)

We pursued the possibility that Slc22a11 (mOat4) might be expressed in mouse RPTECs, encouraged by homologous alignments of Slc22a11 transcripts to mouse chromosome 19 from related rodent species (See Appendix A). Mouse-specific primers were designed from the four-exon predicted Slc22a11 transcript, Gm50435. We tested the total RNA isolated from mouse placenta and kidney, which are the primary organ tissues (*n* = 4 replicates) known for Slc22a11 expression [65,66]. The results of the ddPCR analysis, as shown in Figure 9, obtained with primers crossing the boundaries for exons 2–3, indicated the strongest signal at 10.1 ± 0.3 copies/µL in the kidneys and 2.8 ± 0.5 copies/µL for placenta. Primary cells and the LgT and hTERT cell lines exhibited low but detectable expression in the range of 0.5 to 1.0 copy/µL.

LgT expression was slightly lower (*p* < 0.05) than hTERT cells but not different from parent cells. The expression results, using a primer targeting exons 3–4, showed a lower level of expression but the same proportional differences among cell types, as well as for kidney or placenta, as the results obtained with the other primer. Notably, kidney expression of Slc22a11 was 2- to 4-fold greater than in placenta, depending on the primer.

### 2.9. Nanostring Transcript Analysis

To confirm the RNA-seq expression of transporters, we designed a transcript panel of 52 probes using the multiplex nCounter^®^ analysis system (Nanostring), which covered most of the renal transporters shown in Figure 8. The expression of transcripts by nCounter^®^ was set at a mean of ≥5 fluorescent reads (*n* = 6/cell type). The results presented in Appendix A show that nCounter^®^’s hybridization-based probes agreed well with the RNA-seq data in the detection of the wide expression range of transporter transcripts among the three cell types. We found that nCounter^®^ was slightly more sensitive than RNA-seq in detecting some transcripts with limited expression, including Mate2K, Mrp6, Oat2, Oat5, Oat8, Oatp1a4, and Urat1. Even so, we observed that the RNA-seq detected other low-expression transcripts (Mrp2 and Oat10) not found by the nCounter^®^ platform. Overall, orthogonal analysis using the nCounter^®^ platform supported the RNA-seq data.

### 2.10. Chemically Inducible Renal Transporters

We wanted to determine whether the transcripts from some of the transporters presented in Figure 8 were inducible or downregulated in these two cell lines. To examine this question, we referred to our previous work for which hTERT and LgT cells were repeatedly exposed to CisPt in culture and then followed by RNA-seq, which was performed every 24 h over a span of 24–96 h [44]. Figure 10 shows the fold changes in the expression of 18 transporters appearing in Figure 8. Over half (14 of 18) of the transporters were inducible in both cell lines, with an earlier expression beginning at 24 h in LgT cells. Drug transporters Mdr1, Mdr2, Mrp2, Pgp, Bcrp, and Oat2 increased in expression in response to CisPt in our prior study. (Note that Oat2 was previously detectable in our earlier RNA-seq data.) However, some transcripts were increasingly downregulated from time-matched controls from 24 to 96 h in both cell types and included Oatp4c1, Oatp1a6, Npt2b, and Npt1. Also, hTERT cells showed that Sglt2 and Abca1 transcripts continued to decrease from 24 to 96 h.

### 2.11. Genomic Karyotyping and Copy Number

Chromosomal alterations from the parent cell line associated with immortalization in hTERT and LgT cells were assessed by whole-genomic sequencing (WGS) and RNA-seq of each cell type. WGS was conducted at an approximately 100× sequencing depth of 500–530 million read pairs each for parent, hTERT, and LgT cells that mapped at >95% to GRCm38-mm10. The estimates of the copy number variation (CNV) from the WGS were made using the CNVkit software [67]. Further, we made use of the expression values and B-allele frequencies (BAFs) from the RNA-seq reads with the CaSpER algorithm to estimate the CNV events across the genome [68]. The WGS and RNA-seq results are displayed using a karyotype map in Figure 11. For hTERT cells, the WGS shows the loss of major portions of chromosome 4, which is reflected in the RNA-seq data. Similarly, portions of chromosome 9 were also lost but not as extensively as chromosome 4. The WGS data also show some gains in chromosome 10 in hTERT cells, observed in five out of six RNA-seq samples. For LgT cells, the genomic data primarily indicate the largest increases in the copy number occurred in chromosome 5, as substantiated by increases in the transcriptomic data of each sample. It should be noted that minor variations in the sequencing depth in the RNA-seq data may account for some of the inter-sample differences in the CNVs for the hTERT and LgT cells.

We are interested in the correlation of CNVs with transcriptional changes in hTERT and LgT cells (Appendix A). The heat map in Figure 12A suggests that the chromosomal loss in the hTERT cells was correlated with a decrease in the expression of the 1014 and 741 transcripts in chromosomes 4 and 9, respectively. In these two chromosomes, we found a downward expression of >2-fold for 16 ion channels, 7 Cyps, and 19 Slc transporter genes. On chromosome 4, the sodium transporters Scn2b, Scn3b, Scn4b, and Scn5a were greatly reduced, along with reductions in Trf (transferrin) expression. In addition, we found an increase in the expression of 768 transcripts that correlated with CNV gains on chromosome 10, although the fold changes in the expression were relatively minor, increasing only up to 13-fold. The LgT cells exhibited a different pattern of chromosomal alterations than hTERT cells. We found an increase in 1015 transcripts that correlated with CNV gains for chromosome 5 in LgT cells (Figure 12B). Notable increases in expression occurred with developmental genes, such as Cdx2 (caudal-type homeobox 2) with a 77×-fold increase and Gbx1 (gastrulation brain homeobox 1) with a 7×-fold increase; elevated interferon-stimulated genes—Oas3, Oasl1, Oasl2, Oas1h, Oas2, and Oas1a, with increases ranging from 6×-fold to 32-fold, recognized as responses to a viral LgT antigen [69]; elevated expression of six Slc transporters; and a 5×-fold increase in Brca2. Overall, we observed the predicted changes in copy numbers for hTERT (Chr 4, 9, and 10) and LgT (Chr 5) cells, which correlated well with the gene expression changes.

## 3. Discussion

In this study, we compared the transcriptional and genomic profiles of two immortalized RPTEC cell lines with parent primary cells to better understand their phenotypic responses [44]. Here, we found about 5 to 6000 Ensembl transcriptional changes in hTERT and LgT RPTECs compared with parent proximal tubule cells, which had somewhat more downregulated genes compared to the 1400 to 2100 upregulated genes. When the altered genes were filtered for those with functional annotation, we found that 92.5% of the expressed genes were shared among the parent, hTERT, and LgT cells. This observation suggests a high degree of similarity in gene expression among the primary renal proximal tubule cells and the immortalized hTERT and LgT cell lines. We explored the hypothesis that the differential expression of specific genes and pathways might explain the transcriptional responses of each cell line to aflatoxin and cisplatin exposure in prior work [44], as well as to reflect upon the potential use of these cells lines in future pharmacodynamic and toxicologic studies.

We followed up on transcriptional changes that might affect AFB1 metabolism, given the sensitivity of LgT cells to AFB1 toxicity. Bioactivation of AFB1 depends upon metabolic activation to the exo-8,9-epoxide by specific cytochrome P450 enzymes [60,70]. The presence of Cyp1a1, Cyp1b1, and Cyp3a13 enzymes in both hTERT and LgT cell types suggests that each cell line is capable of activating AFB1 to the toxic, epoxide metabolite. We also examined those mouse transcripts that directly detoxify AFB1 (Akr7a5, Ephx1, Ephx2, Gsta1, Gsta3, Gstk1, Gstm1, Gstm2, and Gstm3) or that might reduce oxidative stress (Cat, Gsr, Gss, Gpx3, Sod1, Sod2, and Sod3) as a consequence of AFB1 metabolism. Our results suggest that increases in GSH S-transferases and epoxide hydrolases [59,71] in hTERT cells over parent and LgT cells would position them to better detoxify AFB1 than LgT cells, which could explain the decreased sensitivity of hTERT cells to AFB1 toxicity compared to the more susceptible LgT cell line. Future AFB1 metabolite analyses should confirm this conclusion.

For CisPt, we previously reported that hTERT and LgT cells both displayed injury and repair transcriptomic responses after repeated low-level CisPt exposure from 3 to 96 h. LgT cells typically showed a more rapid increase and greater number of DEGs from 24–72 h and larger fold increases for highly expressed genes at 100× to >1000× fold compared to hTERT cells in the 25 to 100× fold range of upregulation [44]. To explain these differences in the magnitude of the response, we conducted the current study looking for changes in signaling pathways, Cyps, ion channels, and transporters in the hTERT and LgT cell lines compared to the parent proximal tubule cells. Pathway analysis was most informative in providing evidence for a mild but persistent genotoxicity in LgT cells indicated by low-level fold increases in genes for cell cycle (e.g., Chek2 and Mcm transcripts) and DNA repair pathways (e.g., Brca1,-2, Rfc subunits, and DNA polymerases). We speculate LgT cell’s increased transcriptional responsiveness to CisPt (versus hTERT in previous work) may be partly due to some level of genomic instability caused by p53/Rb-binding from LgT antigen during immortalization [72,73]. Other notable features of the LgT cells from parent RPTECs observed here were the upregulation of Trex2 for replication fork stability [74] and increases in several Hox13 genes (Hoxb13, Hoxa13, and Hoxd13) acting as pioneer factors for genes involved in cell morphogenesis and differentiation that we interpreted as an adaptive response to immortalization [75]. Similar to LgT cells, hTERT cells also showed sizeable increases in the Trex2 fold change compared with parent proximal tubule cells, as well as increased transcripts for morphogenesis factors (e.g., Nkx2–3, Esx1, and Six1) and interferon-beta (Infβ1). Overall, a degree of replication fork stress after immortalization would not be unexpected from either telomere elongation [76] in hTERT cells or p53/Rb binding [77] in LgT cells that would ultimately affect how these immortalized cells react to chemical exposures [73].

The expression of metabolic transporters in renal proximal tubule cells play a crucial role in CisPt chemotherapy [78]. Renal proximal tubule cells are particularly sensitive to CisPt accumulation by its continued import from the influx transporters Oct2 (Slc22a2) and Ctr1 (Slc31a1), which can outpace CisPt elimination, overwhelm biochemical defenses and repair systems, and eventually lead to cell death and renal failure in vivo [79]. Renal proximal tubules are equipped with many efflux transporters that can recognize and eliminate both CisPt and its metabolites into urine. CisPt relevant efflux transporters include Mate1, Mate2K, Mdr1, Mrp2, Bcrp, Atp7a and Atp7b (Slc47a1, Slc47a2, Abcb1b, Abcc2, Abcg2, Atp7a, and Atp7b, respectively) [80,81]. The availability of so many efflux transporters in proximal tubule cells makes CisPt toxicity seem unlikely. However, the balance of CisPt uptake and elimination in a clinic depends on a patient’s variable repertoire of functional renal transporters and their age and disease state, which together can present grave vulnerabilities for adequate renal function during medical treatment. Complicating chemotherapy is that an adaptive response of many tumors is to overexpress efflux transporters, creating CisPt resistance and the need for higher CisPt administration for effective antitumor treatment, placing further burdens on the kidney [78,82]. In our study, a selected panel of transporters (Figure 8) showed there were comparable levels of expression of pharmacodynamic transporters among parent primary cells and moRPTEC cell lines. Ctr1 was highly expressed, and Oct2 expression was relatively low in all cell types. CisPt efflux transporters were highly or moderately expressed in all cell types except for the minimal expression of Mate2K. Although we did not perform CisPt analysis, we expect import and accumulation did occur over time in hTERT and LgT moRPTECs, given each cell type displayed transcriptional profiles of injury and repair [44]. While the correlation of transcript changes with protein expression varies in many cell types, pathway analysis is an effective tool for improving the focus of differential expression and interpreting the high volume of data from transcript profiling [83,84]. We used this approach to gain mechanistic insight into the chemical responses and effects of immortalization in hTERT and LgT cells. Overall, the heightened sensitivity we reported for LgT cells to CisPt (versus hTERT cells) likely stems from their differing adjustments in metabolic, transcriptional, and signaling pathways brought about by replication stress after immortalization [73,85,86].

Immortalized cells in 2D culture retain many properties of primary cells, but the loss of the 3D structure of the organ, pulsatile blood flow, and hormonal support during proximal tubule cell isolation can limit the expression of some genes compared to cells in the intact kidney [87]. For example, Oat’s and Oct’s transporters, generally, have low expression in cultured primary renal cells or renal cell lines compared to the whole organ. Investigators looking to study pharmaceutically important transporters like Oat1 and Oat3 in cultured cells have boosted their expression by viral transduction of these genes into recipient cell lines [88] or have found increased Oat transporter expression in promising new methods like renal tubule spheroids [89]. Even so, there is great interest in standardizing more physiological in vitro RPTEC models for reproducible comparability among research and pharmaceutical labs for drug discovery and toxicity screening. A recent study extensively investigated toxicant effects on barrier formation, directional transport, and gene expression using various human RPTECs cultured on transwell membranes under static and fluidic conditions using the PhysioMimix T12 organ-on-chip system with 2 µL/sec flow [27]. The authors report primary RPTECs were challenging to work with because of cell clumping on the transwell membrane and inconsistent proximal tubule barrier and transport function. They found immortalized TERT1-parent and TERT1-OAT1 cells were more amenable to transwell cultures and their transcriptomes compared more favorably with human kidney medulla than other cells lines tested. Cisplatin or PFOA treatments (perfluorooctanoic acid) produced 257 DEGs in these two cell types representing enrichment in pathways for glycolysis/gluconeogenesis, cell signaling, and hypoxia-inducible response. Recent reviews of in vitro RPTEC models for pharmacokinetic profiling suggest flow-based models help prevent buildup of endogenous byproducts and chemical metabolites to provide a more accurate assessment of gene and pathway expression for xenobiotic activation, conjugation reactions, and transporter activities [90,91].

Here, we found that RNA-seq and nCounter^®^ platforms could detect 5 out of the 10 Oat transcripts and all 3 of the Oct transcripts in moRPTECs. Using the sensitivity of ddPCR, we were also able to detect the mOat4 (Slc22a11) transcript, for the first time, from the kidneys and placenta, as well as low-level expression in RPTECs. Two separate, exon-crossing primers were successfully used in this analysis to confirm Slc22a11’s expression. However, once proximal tubule cells were isolated from whole kidney, ddPCR could only detect low-level expression in our moRPTEC cell lines (≤1 copy/µL). Recently, investigators applied an elegant and sensitive surrogate peptide protein quantitation method using mass spectrometry to measure expression changes in 35 transporters in mice and rat steatohepatitis models, and being unable to detect mOat4, this suggests the gene may not be present in rodents [92]. In human pharmacotherapy, Oat4 is known for its involvement in uric acid transport (e.g., gout) and for its affinity for many therapeutics affecting their pharmacokinetics and disposition [93,94,95]. It is also notable that additional studies suggest the Slc22a11 transporter may be of environmental and public health importance. Slc22a11 activity studies after transfection into HEK (human embryonic kidney) cells suggest many PFAS compounds are recognized by this transporter and could be reabsorbed, thereby contributing to their long half-life in humans [66]. The additional roles of fetal protection from PFAS exposure have been attributed to Slc22a11 activity in an ex vivo recirculating human placental perfusion system [65]. The potential for recognition of PFAS compounds by renal transporters like Oat1–4 and Oct2 from public environmental exposures may have important clinical implications since adjustments may be necessary in calculating the renal clearance of pharmaceuticals in humans [96]. We expect that the cloning of mouse Slc22a11 to define both transcript and peptide sequences will greatly assist future studies in mouse models examining its effects on both therapeutic agents and environmental contaminants.

Looking at our previous work with hTERT and LgT cells, we also found that several transporters, including Mdr1, Mdr2, Mrp2, Pgp, Bcrp, and Oat2, among others, were inducible in culture by CisPt, suggesting retention of signaling processes in these cell lines that can transcriptionally respond to chemical exposure. Generally, we found that the transcripts for Cyps, ion channels, and transporters in the hTERT and LgT cells were well represented compared to primary parent proximal tubule cells. We acknowledge that further studies with moRPTEC cell lines to assess drug and chemical transport in functional assays would be greatly facilitated in matrix cultures or spheroids and tubuloids for preservation of polarized structures and transporter functions [87,89,97].

Epithelial cell immortalization involves a complex process of selection for continued cell division bypassing senescence barriers while maintaining cell-of-origin gene expression, biochemical functions, and contact inhibition [98]. We have described changes in gene expression and regulatory pathways in immortalized hTERT and LgT moRPTECs consistent with these properties. The WGS allowed us to karyotype chromosomal alterations in these cells compared to parent cells, and the results suggest that the copy number variation (CNVs) of genes plays a pivotal role in supporting the selection process toward immortalization. We believe that stability in CNV changes are important for maintaining an immortalized state either by hTERT or LgT antigen expression. Some genetic drift over time may be possible. Telomerase elongation to immortalize cells prevents telomere erosion so that cells do not reach the limit where replicative senescence and apoptosis occur [99]. For hTERT cells, WGS showed loss of major portions of chromosome 4 and parts of chromosome 9, reflected by the downregulation of most transcripts found in these chromosomes, with some copy number gains in chromosome 10 as well. Loss or repression of the cell cycle checkpoint inhibitor Cdkn2a(p16^Ink4a^) is a genomic change often associated with hTERT immortalization [99,100], because Cdkn2a can no longer effectively sequester Cdk4/Cdk6 kinases, thereby allowing continued cell proliferation [101]. Here, we found that the Cdkn2a expression was unchanged in hTERT moRPTECs compared with parent cell levels, while another Cdk inhibitor, cdkn1c(p57^Kip2^), was reduced 20-fold, which is consistent with its reported loss in other immortalized epithelial cells [58,102]. Only annotated genes were analyzed by filtering out all ‘NA’ (non-annotated) genes from the deSEQ2 output. For LgT cells, SV40LgT immortalization occurs by the binding of LgT Ag to Trp53 and Rb proteins, impairing their abilities as negative regulators of the cell cycle so that continued cell division can occur. Again, the resting levels of Cdkn2a were not reduced in LgT cells but instead increased over 10-fold from parent cells. We believe this increase in Cdkn2a levels in LgT cells actually reflects an alternate activity of Cdkn2a (separate from Cdk inhibition) as a response to and protection from DNA damage, as reported in recent research [103,104].

## 4. Materials and Methods

### 4.1. Cell Immortalization and Culture

Mouse renal proximal tubule epithelial cells (moRPTECs) were isolated from CD-1 mice, as previously described in [44], and grown on Biocoat collagen-1-coated 6-well plates (Corning, Corning, NY, USA; cat. no. 356400) in culture medium (ScienCell, Carlsbad, CA, USA; cat. no. EpiCM-a) containing 2% FCS, penicillin/streptomycin antibiotics, and a growth supplement. Although commercial growth supplements are often proprietary, they typically contain EGF (epithelial growth factor), insulin, hydrocortisone, transferrin, and other hormones necessary for epithelial cell growth in culture, as described in other studies [27,105]. Other growth mediums and contact surfaces (e.g., laminin-coated dishes and transwells for polarity; hydrogel microfibers and circulating hollow fiber systems for 3D architecture and flow) may also be successfully adapted for moRPTECs as needed. Parent moRPTECs were transduced with lentivirus vectors containing either human TERT or SV40 Large T (LgT) with antibiotic selection modules, as previously described [44]. Briefly, the Ef1a_Large T-antigen Puro lentivirus (Addgene, Watertown, MA, USA; cat. no. 18922) contained a puromycin selection marker. The lentivirus expressing hTERT under enhanced expression of an EF1a promoter contained a neomycin selection marker (GenTarget Inc., San Diego, CA, USA; cat. no. LVP1131-Neo). Puromycin or Neomycin are required for LgT antigen or hTERT in these vector-driven expression systems, respectively, to maintain a uniform immortalized population. The specific lentivirus vectors confer antibiotic-resistance for selective expression of the desired gene (hTERT or LgT), so that non-vector-containing cells are eliminated. The concentration of each of these antibiotics was carefully determined to prevent cytotoxicity in vector-containing cells while maintaining a propagating immortalized cell type. The process of creating immortalized cell types does lead to some transcriptomic changes, but the proximal tubule transcriptome remains highly intact, since 92.5% of transcripts were maintained in immortalized cells compared with primary parent cells.

Cells were passaged for over 50 cell divisions under continuous antibiotic selection. Parent moRPTECs and LgT and hTERT immortalized moRPTECs were grown to near confluence in 6-well plates. DNA and RNA were harvested separately in separate 6-well plates for either DNA- or RNA-sequencing analysis.

### 4.2. RNA and DNA Isolation: RNA-Seq, DNA-Seq

The cell viscosity was reduced by the use of cell column shredders prior to nucleic acid isolation (Qiagen, Germantown, MD, USA; Qiashredder cat. no. 79654). For the RNA isolation (Qiagen, Germantown, MD, USA; RNeasy Mini, cat. no. 74004), the cells were washed twice with PBS prior to the addition of lysis buffer and spin columns with on-column DNAase-1 digestion followed by washes, elution, and storage at −80 °C. The RNA integrity was measured using an Agilent Model 5300 Fragment Analyzer (Agilent, Santa Clara, CA, USA) with RQN values of 9–10. RNA samples, at *n* = 6 per cell type, were prepared individually for RNA-seq by rRNA depletion, fragmentation (Covaris Inc., Woburn, MA, USA), conversion to cDNA (including mRNA and ncRNA), bar coding, and library construction, as previously described [106], with some modifications. Briefly, pooled libraries were analyzed for the cluster generation of 150 bp paired-end fragments to produce an approximately 40× coverage of the mouse transcriptome using a NovaSeq6000 instrument (Illumina, San Diego, CA, USA). The RNA-seq reads were aligned to mm10 with STAR (spliced transcripts alignment to a reference). The repeat regions were filtered out as described in Section 4.4. The differential expression was determined using DeSeq2 [107]. For genomic analysis, DNA from each cell type was isolated with commercial spin columns (Qiagen DNeasy Kit, cat. no. 69504). Similar to the above description, DNA was fragmented and bar coded, and libraries were prepared for the 150 bp paired-end whole-genome sequencing using an Illumina NovaSeq6000 instrument for an approximately 100× coverage. All raw fastq data files for DNA-seq and RNA-seq are stored in the NCBI Sequence Read Archives (SRA) public database, under Bioproject PRJNA1142928.

### 4.3. Genomic Karyotyping: WGS and RNA-Seq

Paired-end DNA-seq reads were trimmed and quality filtered using the TrimGalore command line utility with flags --stringency 1 --length 40 (TrimGalore, v0.6.6; https://www.bioinformatics.babraham.ac.uk/projects/trim_galore/ (accessed on 23 August 2023)). High-quality trimmed reads were aligned to the mouse genome (mm10) with bwa mem v0.7.15 with the default parameters [108,109]. Mapped reads were extracted and quality filtered using Samtools v1.18 with a view with the flags-F 4-q 20 [110]. The Picard command tools (v21.0.1), SortSam and MarkDuplicates functions, were used to sort the binary alignment files (BAM) and mark duplicates (https://broadinstitute.github.io/picard/ (accessed on 27 August 2023)). The CNVkit pipeline (v0.8.1) was employed to determine the whole-genome copy number variations (CNVs) in hTERT and LgT immortalized cells, considering the moRPTEC parent as the normal reference [67].

### 4.4. Differential Expression by RNA-Seq

Paired-end RNA-seq reads were aligned to mm10 with STAR v2.6.0 [111]. The SAMtools (v1.18; and view function with flags -F 4 -q 20) was used to filter out multimapping reads (reads that align to multiple locations), thus reducing extraneous signals, which occur in repeat regions. The reads with a mapping quality < 20 were discarded. Gene counts were quantified with the featureCounts command line utility, and a gene expression analysis was performed using DeSeq2 in the R statistical programming environment [107,112]. Differential expression was characterized by an absolute fold-change ≥ 2 and Benjamini–Hochberg adjusted *p*-value ≤ 0.05.

### 4.5. Venn Diagram and Pathway Analysis

Differentially expressed genes (DEGs) from the RNA-Seq with ≥2-fold change and pAdj < 0.05 were used as input for the Venn diagram analysis conducted online at the following web portal: https://bioinformatics.psb.ugent.be/webtools/Venn/ (V1; accessed on 22 July 2024). Only annotated genes were analyzed by filtering out all ‘NA’ (non-annotated) genes from the DeSeq2 output. The Venn diagram analysis determined common and uniquely expressed genes for parent, LgT, and hTERT cell types. Common and unique gene sets from the Venn diagrams were used as input into the Ingenuity Pathway Analysis (IPA) software v2024.4 (licensed for use from Ingenuity Systems, https://apps.ingenuity.com/ingsso/login, accessed on 26 May 2024) as previously described [106]. Briefly, the IPA Core Analysis module was used to find the top canonical and disease pathways populated by differential expression. The significance value associated with overrepresented pathways measures the likelihood of an association between an experimental gene set and a reference gene set for a specific process or pathway. The *p*-value is calculated with the right-tailed Fisher’s exact test. Ingenuity (IPA) uses public databases (e.g., HumanCyc) and performs in-house curation to formulate and update signaling pathways and gene transcript and product interactions.

### 4.6. Clustering Analysis

Subsets of genes functionally categorized as either cytochrome P450s, ion channels, or metabolic transporters were selected for closer examination of the expression patterns across the three cell lines. DeSeq2’s normalized expression values were extracted for each functional category and subjected to hierarchical clustering using the R hclust function of the ComplexHeatmap library [113]. Heatmaps were sliced with k-means clustering at km-row = 4, 4, and 7 for Cytochrome P450s, ion channels, and transporters, respectively.

### 4.7. CNVs

The CaSpER pipeline was employed to detect CNVs from the RNA-Seq data [68]. In brief, B-allele frequencies were extracted for all samples using the BAFExtract command line tool. Segmentation and CNV event prediction were conducted and summaries generated for hTERT and LgT against the moRPTEC reference, with CaSpER in R. The multi-sample CNV intersections were derived using bedtools v2.25.0 multiIntersectBed function for the hTERT and LgT immortalized cell lines [114]. The high-confidence CNV set for each cell line consisted of CNV segments present in at least five of the six samples. The RNA-Seq and WGS CNVs were visualized using the R karyoploteR library, with modification for the mm10 genome [115].

### 4.8. Nanostring Analysis

The NanoString’s nCounter^®^ (NanoString, Seattle, WA, USA) platform was used as a multiplexed gene expression platform to test for specific transcripts, as previously described [116]. Briefly, non-cross-reactive probes were custom synthesized by Nanostring and mixed to validate transcript expression after normalization with housekeeping transcripts (Actb; Gapdh, Hprt, and Rpl32), which were determined by RNA-seq to be treatment stable. Samples were analyzed at 150 ng RNA input.

### 4.9. ddPCR Analysis for Slc22a11

A total RNA amount of 1000 ng was used for the cDNA synthesis, which was reverse transcribed using an iScript cDNA synthesis kit (Bio-Rad, Hercules, CA, USA) following the manufacturer’s recommendations. The PCR reaction mixture was assembled from the ddPCR Multiplex Supermix (Bio-Rad), DTT 4 mM, probe mix (final 250 nM concentration, for each probe), and 5 μL of the cDNA template in a final volume of 25 μL, following the manufacturer’s instructions (Bio-Rad). Twenty microliters of each reaction mix were converted to droplets with an Automated Droplet Generator (Bio-Rad). Droplet-partitioned samples were then transferred to a new 96-well plate, sealed with a PX1 PCR plate sealer, and cycled in a T100 Thermal Cycler (Bio-Rad) under the following cycling protocol: 95 °C for 10 min (polymerase activation), followed by 40 cycles of 95 °C for 30 s (denaturation), 60 °C for 2 min (annealing), and a final 98 °C for 10 min (polymerase deactivation) and an infinite 4 °C hold. The plate was transferred and read in the HEX, Cy5, and ROX channels using the QX600 reader (Bio-Rad). QX Manager Software 2.1 Standard Edition Bio-Rad estimated the number of template molecules per microliter of starting material. The samples were normalized using an Tbp (TATA-binding protein) probe for all samples (Table 3). The normalization factor was derived by taking the average of Tbp across all wells in the plate and dividing the individual wells by the average. The normalized concentration for each transcript was determined by dividing the concentration by the normalization factor and plotted as the normalized concentration (copies/µL).

### 4.10. Statistical Analysis

nCounter^®^ multiplexed data (Nanostring) and BMC (base mean count) data for the scatterplot analysis from the RNA-seq and ddPCR data were each analyzed by ANOVA and then by Dunnett’s or Newman–Keuls post hoc testing at *p* ≤ 0.05.

## 5. Conclusions

We have created two mouse immortalized RPTEC cell lines, hTERT and LgT, with similar but distinctive transcriptomes compared to the primary proximal tubule cells from which they were derived. Transcriptome profiles including pharmacodynamic and toxicologic genes of interest, such as Cyps, ion channels, and metabolic transporters, show adjustments in expression after immortalization compared to parent cells. Overall, the complement of expressed genes in these cell lines is consistent with renal proximal tubule cells. However, hTERT cells may be more like parent cells compared to LgT cells, since their transcriptional differences and pathway alterations were not as pronounced as LgT cells. The two cell lines responded somewhat differently to chemical challenge, with LgT cells being more transcriptionally responsive than hTERT cells. This may be due to different degrees of replication stress manifest in each cell line from their unique immortalization and selection processes that lead to a stable cell line. Transcriptional and genomic karyotyping would be informative data for any immortalized cell line that might be described as ‘normal’ or ‘like primary cells’, just to be clear about the consequences of immortalization, as well as the expected functional capabilities and potential usefulness in pharmacodynamics, response to chemical challenge, and drug discovery. The value in LgT cells is that they represent renal cells under pathological stress that provide distinct pharmacological responses compared to hTERT cells. The functional properties of moRPTECs in 2D culture and their development into more sophisticated in vitro platforms that exploit 3D architectures, along with different media conditions, await further investigation. In summary, both cell lines bring advantages and unique capabilities to in vitro pharmacological and toxicological chemical screening, which should be translatable into mouse models of kidney disease and therapeutics.

## Figures and Tables

**Figure 1 ijms-26-03607-f001:**
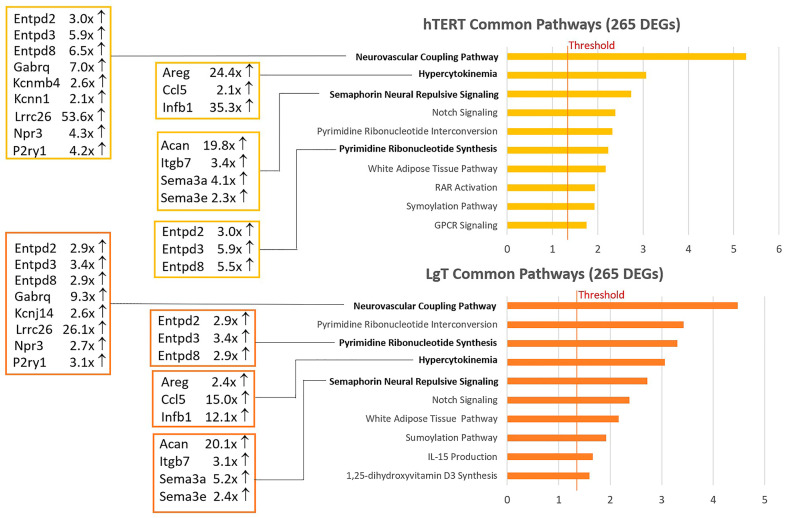
Pathway analysis of common upregulated DEGs from a Venn diagram comparison of hTERT and LgT cells. hTERT or LgT RNA-seq transcripts were compared to primary proximal tubule cell transcript levels to identify differentially expressed genes (DEGs) with pAdj ≤ 0.05, 2-fold change, and ≥5 BMCs (base mean counts). There were 265 upregulated DEGs that were in common between hTERT and LgT cells. These upregulated DEGs were analyzed by IPA pathway analysis. The Z-score is on the x-axis and the top ten rank-ordered pathways are on the y-axis for each cell line. The fold change (compared to primary proximal tubule cells) for select transcripts within bolded pathways are displayed for each cell type.

**Figure 2 ijms-26-03607-f002:**
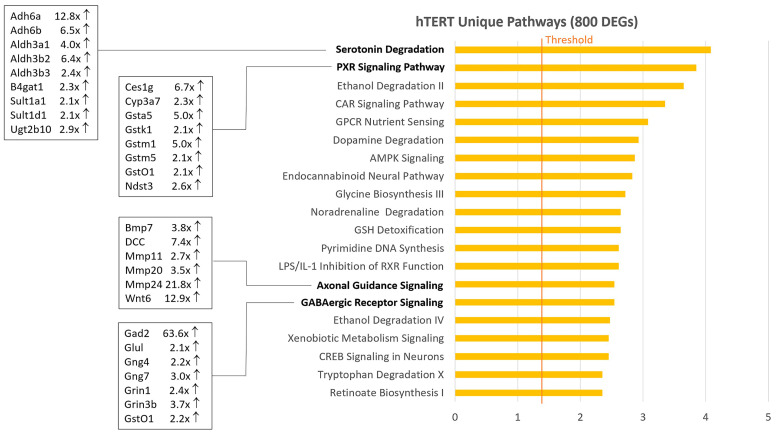
Pathway analysis of 800 unique upregulated DEGs in hTERT cells from the Venn diagram analysis. Differentially expressed genes (DEGs) were calculated as pAdj ≤ 0.05, 2-fold change, and ≥5 BMCs (base mean counts). hTERT-upregulated DEGs were analyzed by IPA pathway analysis. The Z-score is on the x-axis and top twenty rank-ordered pathways are on the y-axis for hTERT cells. The fold change (compared to primary proximal tubule cells) for select transcripts within bolded pathways are displayed for each cell type.

**Figure 3 ijms-26-03607-f003:**
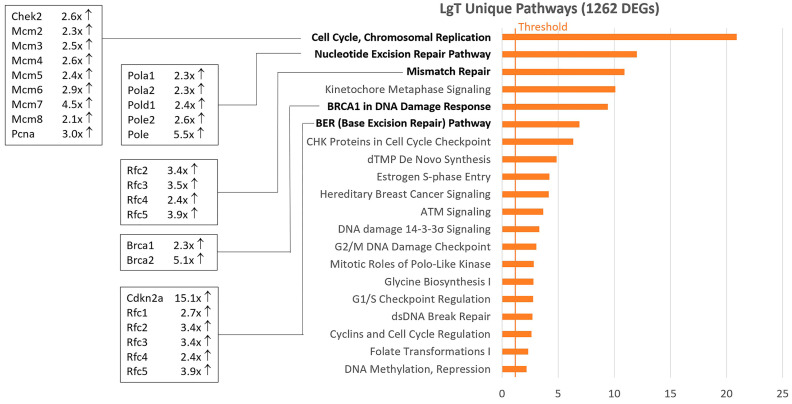
Pathway analysis of 1262 unique upregulated DEGs in LgT cells from the Venn diagram analysis. Differentially expressed genes (DEGs) were calculated as pAdj ≤ 0.05, 2-fold change, and ≥5 BMCs (base mean counts). LgT-upregulated DEGs were analyzed by IPA pathway analysis. The Z-score is on the x-axis, and the top twenty rank-ordered pathways are on the y-axis for hTERT cells. The fold change (compared to primary proximal tubule cells) for select transcripts within bolded pathways are displayed for each cell type.

**Figure 4 ijms-26-03607-f004:**
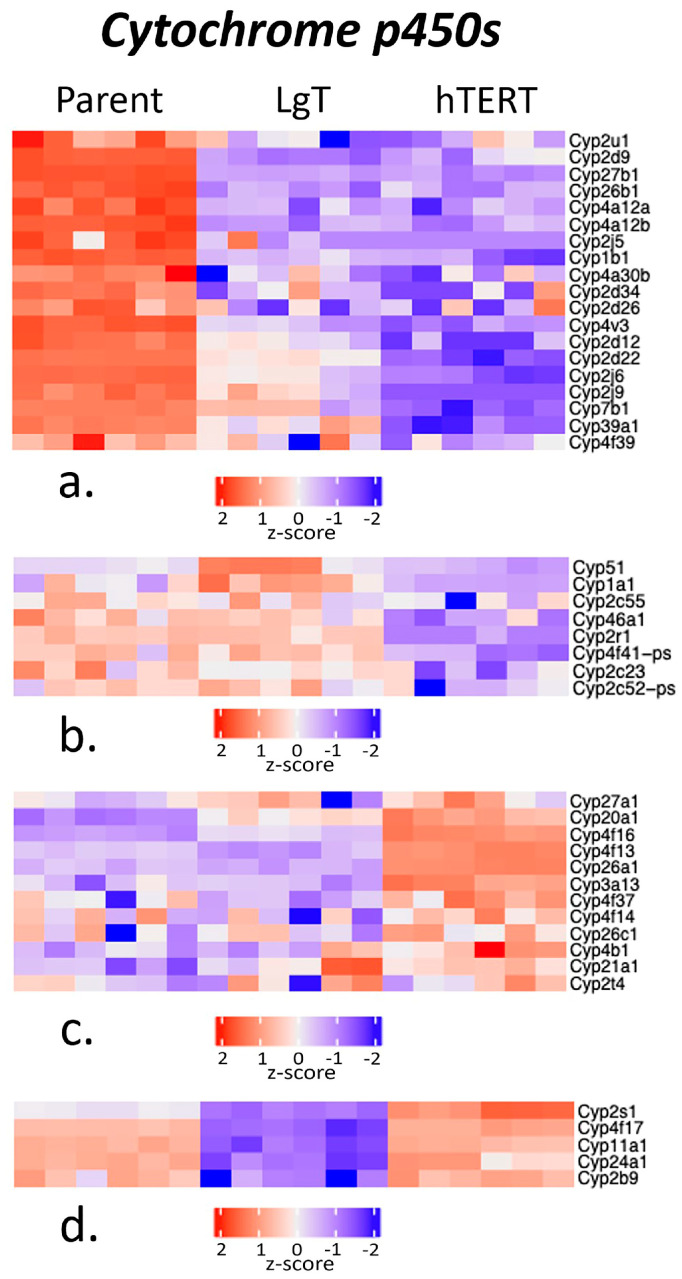
Cluster analysis of the differences in cytochrome P450 (Cyp) expression among parent, LgT, and hTERT cells. The RNA-seq base mean counts (BMCs) for each transcript (*n* = 6/cell type) are compared by Z-score (value range provided in the legend). Hierarchical analysis was used to separate the data into four clusters (**a**–**d**). See text for detailed description.

**Figure 5 ijms-26-03607-f005:**
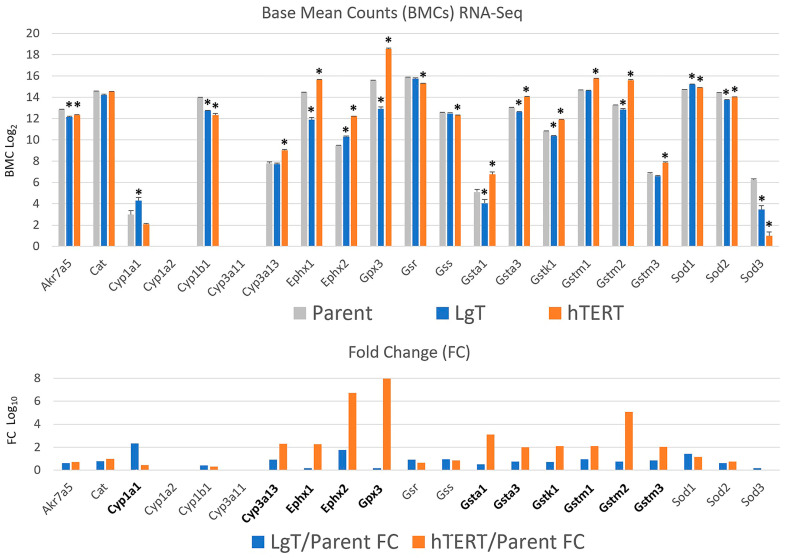
Bioactivation and detoxification transcripts for AFB1 in mouse RPTECs. Relevant ABF1 metabolism transcripts in mice have been previously described (see Section 3) and their relative expression levels in base mean counts (BMCs) are shown in the top panel. Base mean counts (mean ± S.E.M.) are shown in the top panel for the primary parent cells and LgT or hTERT cells at near confluence. Significant differences (*p* ≤ 0.05) from parent RPTECs are indicated by an asterisk (ANOVA and Dunnett’s post hoc test). A log_2_ scale was used to accommodate high- and low-expression transcripts. Note that Cyp1a2 and Cyp3a11 were not expressed in RPTECs. The lower panel shows transcript fold changes in LgT or hTERT cells compared to primary parent RPTECs. Bolded transcripts highlight those with a ≥2-fold change in hTERT (orange bars) or LgT RPTECs (blue bars) relative to parent RPTECs.

**Figure 6 ijms-26-03607-f006:**
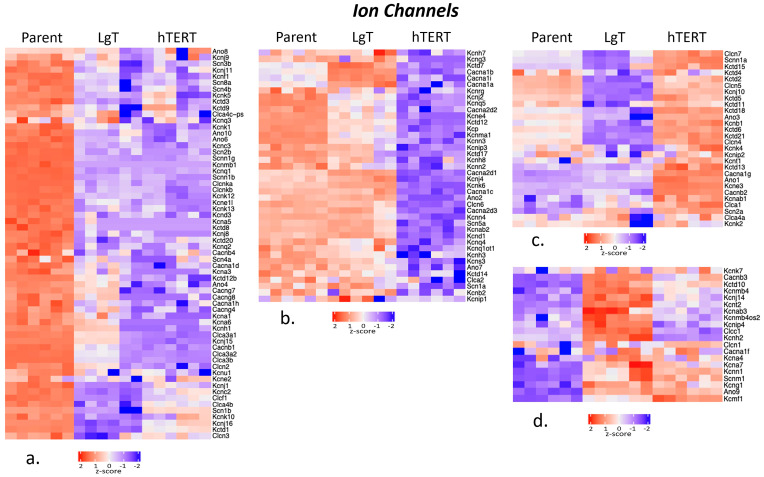
Cluster analysis of the differences in expression among ion channel transcripts in parent, LgT, and hTERT cells. The RNA-seq base mean counts (BMCs) for each transcript (*n* = 6/cell type) are compared by Z-score (value range provided in the legend). Hierarchical analysis was used to separate the data into four clusters (**a**–**d**). See text and supplemental data tables for a detailed description.

**Figure 7 ijms-26-03607-f007:**
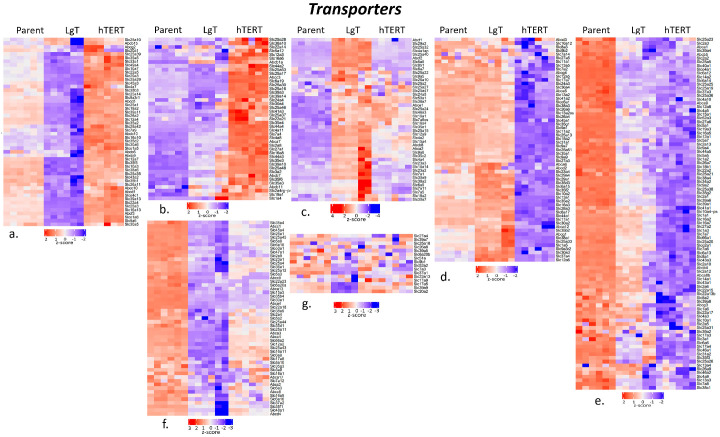
Cluster analysis of the differences in expression among metabolic transporter transcripts in parent, LgT, and hTERT cells. The RNA-seq base mean counts (BMCs) for each transcript (*n* = 6/cell type) are compared by Z-score (value range presented in the legend). Hierarchical analysis was used to separate data into seven clusters (**a**–**g**). See text and supplemental data tables for a detailed description.

**Figure 8 ijms-26-03607-f008:**
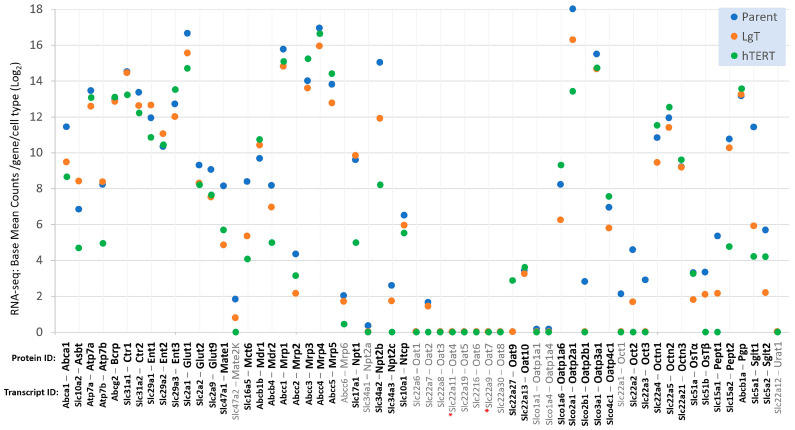
Scatterplot of the transporter expression panel. The base mean counts (BMCs) per gene transcript for the mean of each cell type (*n* = 6/group) are displayed for a transporter panel of pharmacological and toxicological interest. The protein and transcript symbols are shown. Bolded transcripts indicate detection at ≥5 BMCs for at least one or more cell types, while grey transcripts indicate <5 BMCs or below the limit of detection. Transcripts with a red asterisk have not yet achieved a consensus Ensembl identity. See text for details.

**Figure 9 ijms-26-03607-f009:**
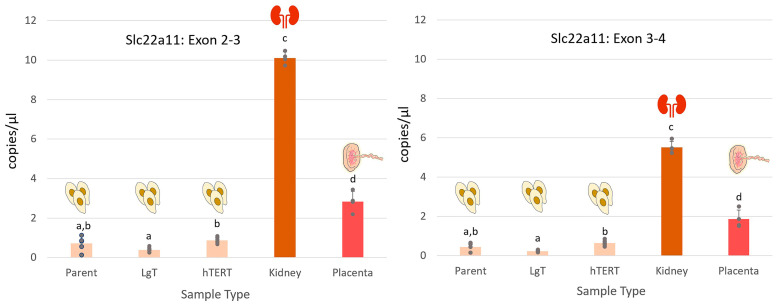
Slc22a11 transporter expression in moRPTECs and tissues. ddPCR was performed using cross-exon primers designed from a partial transcript sequence, Gm50435, predicted for the mouse Slc22a11 gene. Bar graphs show the mean ± S.E.M. of comparable copies/µL with individual data points from identical cDNA input into the PCR reaction from parent renal proximal tubule cells, immortalized LgT and hTERT cells (*n* = 6), and RNA isolated from female kidney and placenta organs (*n* = 4). The sample’s means not sharing a similar letter are significantly different (*p* ≤ 0.05, ANOVA, Newman–Keuls).

**Figure 10 ijms-26-03607-f010:**
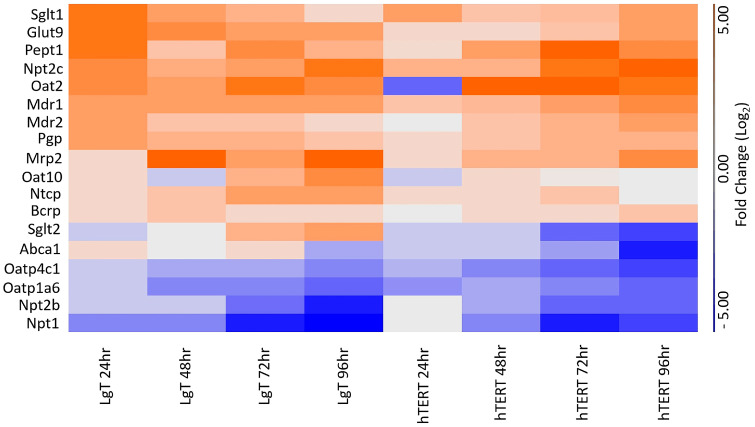
CisPt’s effects on transporter expression over time in LgT and hTERT cells. Data from prior work (Ref. [44]) show fold changes in transporter expression compared to time-matched controls. The fold changes were calculated after repeated CisPt exposure in LgT and hTERT cells. Data for Figure 10 are provided in Appendix A.

**Figure 11 ijms-26-03607-f011:**
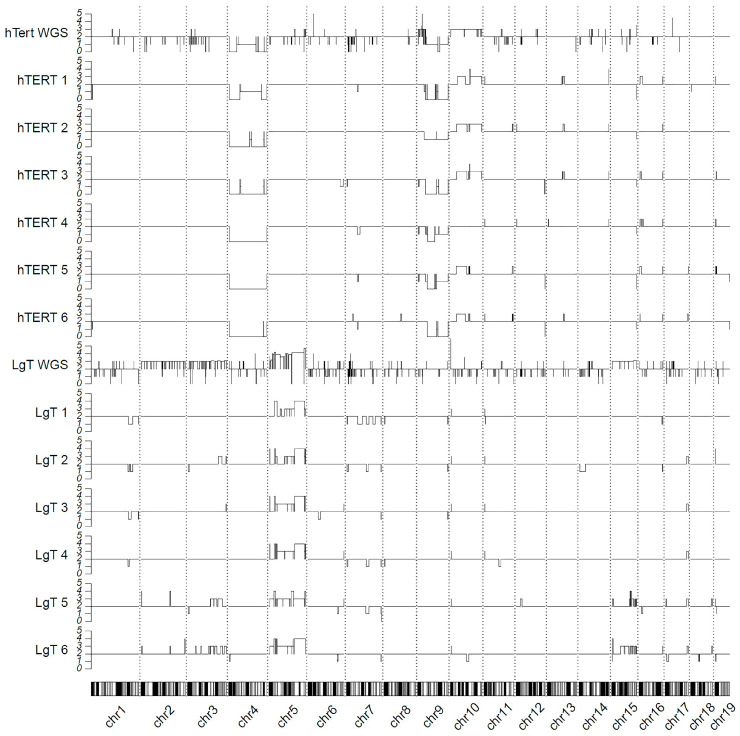
Genomic karyotype map of the hTERT and LgT cells. Whole-genomic sequencing (WGS) and RNA-seq were performed on each cell type. The CNVkit algorithm was used to call copy number variants (CNVs) from the hTERT WGS and LgT WGS. The CaSpER algorithm used RNA-seq reads to estimate CNV events based on transcript expression counts across the mouse genome for each hTERT or LgT sample (*n* = 6/cell type). The y-axis represents the normal allele frequency at a horizontal baseline of 2 (diploid), where the copy number gains are >2 and losses are <2. Chromosomes 1 to 19 were standardized to equal lengths for visual comparison.

**Figure 12 ijms-26-03607-f012:**
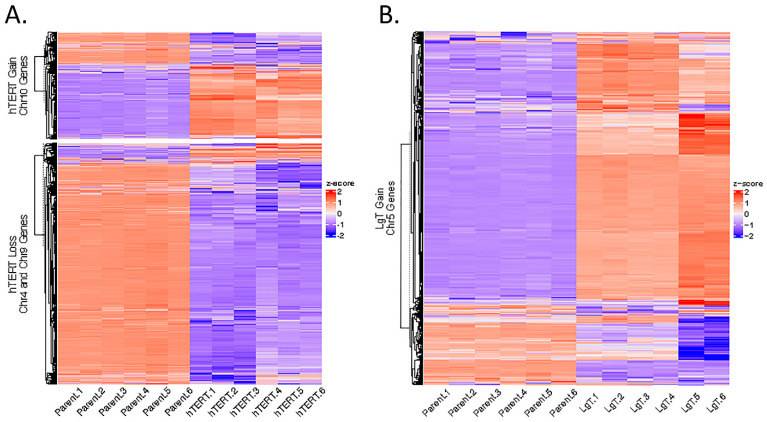
Cluster analysis of the expression differences for transcripts with chromosomal copy number variants (CNVs). Differential expression in hTERT and LgT cells from parent cells was determined for genes appearing on chromosomes 4, 9, and 10 in hTERT cells, as shown in Panel (**A**), and for genes on chromosome 5 in LgT cells, as shown in Panel (**B**). The base mean counts (BMCs) from the RNA-seq analysis for each transcript (*n* = 6/cell type) were compared by Z-score. The range in Z-score values is shown in the legend, located to the right of each panel. A hierarchical analysis was used to separate the expression data into seven clusters a., b., c., d., e., f., and g. See text for a detailed description.

**Table 1 ijms-26-03607-t001:** Total transcripts and differential expression of hTERT and LgT moRPTECs compared to primary parent mouse proximal tubule cells.

Cell Type	Total Transcripts ^1^	Transcripts ^2^ <5 BMCs	Transcripts ^2^ ≥5 BMCs	Total DEGs ^3^	Up DEGs ^3^	Down DEGs ^3^	No Change ^4^
hTERT	55,401	32,027	23,374	6426	1460	4966	9998
LgT	55,401	31,813	23,588	5740	2138	3602	12,333

^1^ Total Ensembl transcripts by RNA-seq. ^2^ Transcripts with either <5 BMCs or ≥5 BMCs (base mean counts). ^3^ Number of upregulated (Up) or downregulated (Down) differentially expressed genes (DEGs) compared to primary moRPTECs that were significant with pAdj < 0.05; BMCs > 5; and 2× fold change. ^4^ Number of DEGs with a <2× fold change but still with pAdj < 0.05 and BMCs > 5 in expression, since they did not meet the 2-fold change threshold.

**Table 2 ijms-26-03607-t002:** Number of metabolic transporter transcripts segregated into 7 clusters.

Transporter Cluster	Total No. Transcripts in Cluster	Order from High to Low Expression per Cluster	No. Transcripts < 5 BMCs		
			Parent	LgT	hTERT
a.	52	hTERT > Parent > LgT	0	0	0
b.	43	hTERT > Parent & LgT	3	2	0
c.	45	LgT > hTERT > Parent	1	0	1
d.	61	Parent & LgT > hTERT	1	1	11
e.	94	Parent > LgT & hTERT	0	6	24
f.	53	Parent > hTERT > LgT	0	7	3
g.	16	Parent = LgT = hTERT	0	1	2
Total	364		5	17	41

**Table 3 ijms-26-03607-t003:** ddPCR probes for mouse Slc22a11 (mOat4).

Gene Symbol ^a^	Assay ID	Probe Fluorophore	Species
Exons 2–3	dCNS754377321	Hex	Mouse
Exons 3–4	dCNS215834387	Cy 5	Mouse
Tbp	dMmuGEXS140712005	Rox	Mouse

^a^ Custom probes for mouse Slc22a11 were designed from the predicted 4-exon transcript, Gm50435, from the Havana Group (Sanger Institute, Cambridge UK).

## Data Availability

Raw sequencing Fastq data files and project metadata have been deposited in the Sequence Read Archives public database under Bioproject PRJNA1142928.

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
