# Peer review of "hTERT and SV40LgT Renal Cell Lines Adjust Their Transcriptional Responses After Copy Number Changes from the Parent Proximal Tubule Cells"

_ijms, 2025, doi:10.3390/ijms26083607_

Round 1
Reviewer 1 Report
Comments and Suggestions for Authors
While there is great interest to have immortalized cell lines that have a clear renal tubular cell phenotype, the approach presented is not complete. A key element of developing the specific phenotype of differentiated tublular cells with clear tubular functional cell properties is not only the initial differentiation but also the maintenance of the specific differential cell state due to the function in the organ context. Thus although a lot of work has been put in comparing gene expression of parantel and immortalized cell lines, the stud fails to present sufficiently funtional renal phenotypes.
A positive notion is the consideration of the cultivation conditions (defined tubulr cell medium with reduced FCS but I miss a discussion why this EpiCM reflects best the needs of tubular cells.
The major short comming is that cells are cultivated on a simple collagen coat in standard 6 well tissue culture plats. In this conditions required cell interactions and mechanical stress needed to mainain the phenotype are not provided.
The other critical point is that gene expression only on RNA level is not sufficient to provide an information about the phenotype. In tubular cells there is an orientation with respect to the membrane and not all mRNA is necessarily translated into protein. Thus while some genes match on the mRNA level there is not demonstration that the immortalized cells obtained actually provide phenotypic cell properties.
Positive comment: I am impressed with the amount of data, but because the conditions are not suited for tubular epithelia functionality, the results are of limited use.
It would be nice to establish a more functional cell cultivation condition and then demonstrate that the immortalized cells have the capacity to readjust their properties to a functional status found in tubular epithelial cells in kidneys in vivo.
Author Response
Reviewer No. 1 Comments and Suggestions for Authors
Comment 1. While there is great interest to have immortalized cell lines that have a clear renal tubular cell phenotype, the approach presented is not complete. A key element of developing the specific phenotype of differentiated tublular cells with clear tubular functional cell properties is not only the initial differentiation but also the maintenance of the specific differential cell state due to the function in the organ context. Thus although a lot of work has been put in comparing gene expression of parantel and immortalized cell lines, the stud fails to present sufficiently funtional renal phenotypes.
Response 1 : We thank the Reviewer for this comment about the range of sophistication of proximal tubule cell models and their applications to research. The Reviewer rightly points out there can be many structural and functional characteristics (e.g. basement membrane, polarity, transporter activity) that could be accounted for in proximal tubule in vitro models. Even so, 3D and microphysiological models do not completely attain renal capabilities observed in vivo. There are still many human and mouse proximal tubule cell 2D models similar to what we describe that are widely used in cell biology, pharmacotherapeutics and cancer research. The revised manuscript now includes for the reader a further discussion of the functional and phenotypic aspects possible for in vitro proximal tubule models depending upon research goals. In addition, we do feel there is novelty in the data presented here for these mouse proximal tubule cell models. Briefly, 1) this is the first hTERT immortalized mouse proximal tubule cell type; 2) these two cell types (hTERT and LgT) respond differently to chemical challenge and we provide an explanation why (e.g. replication stress in LgT cells); 3) we provide evidence for the first time that an important transporter transcript, Slc22a11, is expressed in mice; 4) we provide insights into immortalization related to CNV from genomic alterations; 5) we describe a novel method for genomic karyotyping that quantitatively links transcriptional expression and genomic sequencing.
We address this issue two-fold: in Section1. Introduction in paragraph 3, and in Section 3. Discussion in paragraph 5.
Section 1. Introduction: “Inclusion of continuous or pulsatile flow, polarity, microstructures (e.g. microvilli and basement membrane formation), multicellular organoids, and kidney-on-a-chip are among the many recent features being introduced by research consortia into sophisticated, microphysiological models using immortalized renal cell types [27, 31-34].” Note new references 23, 27, 31-34.
Section 3. Discussion: “Even so, there is great interest in standardizing more physiologic in vitro RPTEC models for reproducible comparability among research and pharmaceutical labs for drug discovery and toxicity screening. A recent study extensively investigated toxicant effects on barrier formation, directional transport and gene expression using various human RPTECs cultured on transwell membranes under static and fluidic conditions using the PhysioMimix T12 organ-on-chip system with 2 µl/sec flow [27]. The authors report primary RPTEC cells were challenging to work with due to cell clumping on the transwell membrane and inconsistent proximal tubule barrier and transport function. They found immortalized TERT1-parent and TERT1-OAT1 cells were more amenable to transwell cultures and their transcriptomes compared more favorably with human kidney medulla than other cells lines tested. Cisplatin or PFOA treatments (perfluorooctanoic acid) produced 257 DEGs in these two cell types and static cultures produced the higher numbers of DEGs versus flow cultures. These DEGs were generally enriched in pathways for glycolysis/gluconeogenesis, cell signaling and hypoxia-inducible response. They concluded that depending on the context of use, immortalized RPTEC cell lines in static transwell cultures may be an appropriate in vitro model.”
Comment 2. A positive notion is the consideration of the cultivation conditions (defined tubulr cell medium with reduced FCS but I miss a discussion why this EpiCM reflects best the needs of tubular cells.
Response 2. We appreciate the Reviewer raising this concern. EpiCM is the commercial product from ScienCell company that they recommend for mouse proximal tubule cells. Since we started with their parent mouse proximal tubule cells and their protocol using EpiCM growth medium, we found both the primary cells and also the immortalized cell lines grew consistently well on a recommended collagen surface.
We have added the following text in Section 4.1 of Material and Methods to address this issue.
“Although the commercial growth supplements are often proprietary, they typically contain EGF (epithelial growth factor), insulin, hydrocortisone, transferrin and other hormones necessary for epithelial cell growth in culture as described in other studies [27, 102]. Other growth mediums and contact surfaces (e.g. laminin-coated dishes, transwells for polarity; hydrogel microfibers and circulating hollow fiber systems for 3D architecture and flow) may also be successfully adapted for moRPTEC cells as needed.”
Comment 3. The major short comming is that cells are cultivated on a simple collagen coat in standard 6 well tissue culture plats. In this conditions required cell interactions and mechanical stress needed to mainain the phenotype are not provided.
Response 3. We thank the Reviewer for bringing up this point. Proximal tubule cells can be cultivated on many different surfaces depending upon intent of the study. The reviewer suggests that cell interactions and mechanical stress are needed to properly maintain proximal tubule cells. For the purposes of our study, we wanted culture conditions that were suitable for cell adherence, propagation and chemical treatment, which we published in our original paper (PMID: 37762531, Ref 43).
We address this point in Section 4.1 of Materials and Methods about culture conditions directed at different study needs and aims when using proximal tubule cells.
“We anticipate other growth mediums and contact surfaces (e.g. laminin-coated dishes, transwells for polarity; hydrogel microfibers and circulating hollow fiber systems for 3D architecture and flow) may also be successfully adapted for moRPTEC cells as needed for different research objectives.”
Comment 4. The other critical point is that gene expression only on RNA level is not sufficient to provide an information about the phenotype. In tubular cells there is an orientation with respect to the membrane and not all mRNA is necessarily translated into protein. Thus while some genes match on the mRNA level there is not demonstration that the immortalized cells obtained actually provide phenotypic cell properties.
Response 4. We appreciate the Reviewer’s concern that transcriptome changes may not completely translate into protein expression that reflects the phenotype of polarized cells. Our aim here was to provide further explanation of results from our first paper (PMID: 37762531, Ref 43) and to provide further insight into the transcriptomes of hTERT and LgT cells in response to chemical challenge, related to their immortalization. While we acknowledge there might be expression differences related to structure/function of proximal tubules, we believe the current cell types do respond like proximal tubule cells to cisplatin and aflatoxin B1 in their transcriptional output.
We address this important point in a two-fold response, in Section 3 of Discussion, Paragraph 4; and in the Section 5. Conclusion for future work.
Section 3, Discussion, Parag 4: “While correlation of transcript change to protein expression varies in many cell types, pathway analysis is an effective tool for improving the focus of differential expression and interpreting the high data volume from transcript profiling [82, 83]. We used this approach to gain mechanistic insight into chemical response and effects of immortalization in hTERT and LgT cells.”
Section 5. Conclusion: “The functional properties of moRPTEC cells in 2D culture and more sophisticated vitro platforms that exploit 3D architectures along with different media conditions await further investigation.”
Comment 5. Positive comment: I am impressed with the amount of data, but because the conditions are not suited for tubular epithelia functionality, the results are of limited use.
It would be nice to establish a more functional cell cultivation condition and then demonstrate that the immortalized cells have the capacity to readjust their properties to a functional status found in tubular epithelial cells in kidneys in vivo.
Response 5. We thank the Reviewer for this comment and their foresight in experiments that remain to be conducted with these cells. As mentioned above, many labs continue to employ 2D proximal tubule cultures for a variety of research objectives.
We comment on this point in Section 5. Conclusion.
“The functional properties of moRPTEC cells in 2D culture and their development into more sophisticated in vitro platforms that exploit 3D architectures along with different media conditions await further investigation.”
Reviewer 2 Report
Comments and Suggestions for Authors
Reviewer opinion
- Is TopHat(Baseed on Bow tie) a good alignment tool for repeated sequence? Are those gene/transcripts with known repeated sequences been checked in this article?
- In section 2.11. Genomic karyotyping and Copy Number: Is the CNVs originate from parent cells, or from the immortalization process? If it’s from the immortalization process, is it stabilized at certain timeframe, or will it keep propagating and changing over time?
- In 4.1. Cell Immortalization and Culture: Why use Puromycin and Neomycin selection markers on hTERT and LgT cells respectively? Will they cause different DEGs profiles?
- In 4.2. RNA and DNA Isolation; RNA-seq, DNA-seq: In references 94, oligo(dT) seems to be used as primer, so non-coding RNA was not included in this study?
- Maybe it’s a dumb question, I am curious whether hTERT RNA transcripts were upregulated in SV40LgT cell in this study? If it’s not upregulated, how did the LgT cell escape the Hayflick limit?
Author Response
REVIEWER No. 2 Comments and Suggestions for Authors
Comment 1. Is TopHat(Baseed on Bow tie) a good alignment tool for repeated sequence? Are those gene/transcripts with known repeated sequences been checked in this article?
Response 1. We thank the Reviewer for this question. In this study, we used the STAR alignment tool to map RNA-seq reads to mouse genes. We appreciate the opportunity to clarify use of this RNA-seq aligner and respond with text in Section 4.2 in Methods.
We revised the text in Section 4.2 of Materials and Methods
“RNA-seq reads were aligned to mm10 with STAR (Spliced Transcripts Alignment to a Reference). Repeat regions were filtered out as described in Section 4.4. Differential expression was determined using DeSeq2 [104].”
Comment 2. In section 2.11. Genomic karyotyping and Copy Number: Is the CNVs originate from parent cells, or from the immortalization process? If it’s from the immortalization process, is it stabilized at certain timeframe, or will it keep propagating and changing over time?
Response 2. We propose that CNV’s originate from the immortalization process. The parent cells were used as the comparator for 2 copies of each gene.
We include revised text below in final paragraph of Section 3 in the Discussion.
“We believe stability of the CNV changes are important to maintaining the immortalized state by either by hTERT or LgT antigen expression. Some genetic drift over time may be possible.”
Comment 3. In 4.1. Cell Immortalization and Culture: Why use Puromycin and Neomycin selection markers on hTERT and LgT cells respectively? Will they cause different DEGs profiles?
Response 3. We appreciate the Reviewer bringing up this point.
We include revised text below in Section 4.1 of Materials and Methods to address to question.
“Puromycin or Neomycin are required for LgT antigen or hTERT in these vector-driven expression systems, respectively, to maintain a uniform immortalized population. The specific lentivirus vectors confer antibiotic-resistance for selective expression of the desired gene (hTERT or LgT), so that non-vector cells are eliminated. The concentration of each of these antibiotics was carefully determined to prevent cytotoxicity in vector-containing cells while maintaining a propagating immortalized cell type. The process of creating immortalized cell types does lead to some transcriptomic changes but the proximal tubule transcriptome remains highly intact since 92.5% of transcripts were maintained in immortalized cells compared to primary parent cells.”
Comment 4. In 4.2. RNA and DNA Isolation; RNA-seq, DNA-seq: In references 94, oligo(dT) seems to be used as primer, so non-coding RNA was not included in this study?
Response 4. We thank the reviewer for this question which we now better describe in the Methods text. Briefly, each sample that was isolated for total RNA was subjected to rRNA depletion. The remaining RNA (includes mRNA and ncRNA) was then converted to cDNA for standard sequencing. ncRNA appears as NA (non-annotated) in Supplemental transcript tables and has an ENSEMBL identifier. However, we chose to focus our work on ‘annotated transcripts’ since these have adequate biological information that is most useful for pathway analysis.
Revised text included in Section 4.2 of Materials and Methods clarifies that mRNA and ncRNA were analyzed by RNA-seq, and we state in Section 4.5 that only annotated genes were included for Pathway analysis.
Section 4.2 – “…conversion to cDNA (includes mRNA and ncRNA), bar coding and library construction as previously described [98] with some modifications.”
Section 4.5 - We state, “Only annotated genes were analyzed by filtering out all ‘NA’ (non-annotated) genes from DeSeq2 output.”
Comment 5. Maybe it’s a dumb question, I am curious whether hTERT RNA transcripts were upregulated in SV40LgT cell in this study? If it’s not upregulated, how did the LgT cell escape the Hayflick limit?
Response 5. We thank the Reviewer for this question. LgT cells do not have hTERT transcripts as their means of continued propagation. LgT cells overcome the Hayflick limit by disrupting cellular senescence pathways from lentivirus-driven expression of LgT antigen that binds to senescence promoting transcripts, primarily - Trp53 and Rb.
Stated in last paragraph (para 8) of the Discussion, Section 3.
“For LgT cells, SV40LgT immortalization occurs by binding of LgT Ag to Trp53 and Rb proteins, impairing their abilities as negative regulators of the cell cycle, so continued cell division can

Round 2
Reviewer 1 Report
Comments and Suggestions for Authors
The quality of the manuscript has improved by adding additional references, comments and discussions regading the very clear limitations of the presented approach. There is however still some omission and may be misunderstanding between the authors regarding my very critical comments. The transwell tert system when stationary is not a significantly improved model. As we and others have shown a key element for improving function of organs that have a high blood flow and other liquid transprot rate improve significantly by enuring a continous transport of medium flow in the system. The high metabolic turn over in RPTEC can only be maintained if medium is continously exchanged. One argument disscussed here that metabolites accumulation in any stationary system directly interfers with gene expression and protein metabolite level control. Interestingly, experiments with single cell types have shown that simple continous (micro)fluidic medium exchange improves the physiological phenotype.
